# Impacts of COVID-19 Restrictions on Young Children’s Outdoor Activity: A Systematic Review

**DOI:** 10.3390/children9101564

**Published:** 2022-10-16

**Authors:** Junjie Liu, Shirley Wyver, Muhammad Chutiyami

**Affiliations:** 1Macquarie School of Education, Macquarie University, Sydney, NSW 2109, Australia; 2School of Nursing and Midwifery, Faculty of Health, University of Technology, Sydney, NSW 2007, Australia

**Keywords:** COVID-19, COVID-19 restrictions, outdoor activity, outdoor play, children, pandemic, lockdown

## Abstract

We sought to identify and synthesise available evidence to aid the understanding of the impact that COVID-19 restrictions had on the outdoor activity of children aged from birth to 12 years. Seven databases (Education Research Complete, ERIC, MEDLINE, PsycINFO, SPORTDiscus, Psychology and Behavioural Sciences Collection, WHO COVID-19 Database) were searched for relevant journal articles in English published from 2020 on. Four qualitative and eleven quantitative studies were included after screening. JBI’s Critical Appraisal Tools were used for quality assessment. All qualitative studies showed an increase in outdoor activity. Less than half of the quantitative studies indicated an increase. Family demography, home characteristics, access to outdoor spaces, and parental support/encouragement/co-play were influential factors. The evidence also supports the recommendation for educators to increase children’s outdoor play time to adhere to the physical distancing guidance and sanitisation requirements. Limited evidence suggests that when COVID-19 restrictions led to decreased outdoor activity, it was associated with less peer socialisation. We identified significant gaps in understanding of the impact of COVID-19 restrictions on young children’s outdoor activity.

## 1. Introduction

The World Health Organization declared COVID-19 a pandemic on 11 March 2020 [1]. As this disease is highly contagious, governments worldwide have advocated for or enforced a range of behavioural interventions to minimise the spread of the disease through social contact. Amongst these interventions is the requirement for people to be confined to their homes [2]. The closing of schools, public venues such as parks, playgrounds, natural areas, and recreational facilities further curtailed possibilities for children to engage in healthful outdoor activity, relieve stress through physical activity, and maintain their physical health [3,4,5]. By restricting access to the outdoors, COVID-19 has abruptly changed children’s outdoor experiences [6]. The outdoors provides dynamic, open-ended opportunities that at times can be unpredictable and risky [7]. As children learn to avoid injury, being outside offers an important opportunity to develop their perception and assessment of risks whilst developing social competence, supporting playful interactions, promoting their social and emotional development, and mastering a wide range of gross motor skills [8,9,10]. During the COVID-19 pandemic, extended school closures and home confinement may have adversely affected young children’s lifestyle behaviours, such as outdoor play, sedentary behaviour, psychology, and mental health [11,12]. Nevertheless, young children may need more opportunities for outdoor physical activity and play in the context of COVID-19 to help them cope with uncertainty and alleviate stress [13,14].

During COVID-19 outbreaks, most countries have taken measures to reduce social contact. These measures have included stay-at-home restrictions, limits to out-of-home activity, closures of schools, parks, and public venues, social distancing, and avoidance of physical contact between people. The duration and severity of these restrictions vary considerably between and within countries [14,15]. Restrictions resulted in limited opportunities for outdoor play and physical activity, although some environmental features have been associated with increased physical activity in some children and youth [15]. Some studies have explored the psychological or mental health effects of COVID-19 on children from different perspectives, such as COVID-19-related fears and their effects on mental health (anxiety and depressive symptoms), school closure, or quarantine impacts on children’s and adolescents’ emotions [16,17,18]. Few comprehensive reviews are available of information on what impacts the pandemic has on children’s outdoor activity. The available studies have investigated physical activity and focused on adults/older adults or children and adolescents [19,20,21]. These studies indicated a decrease in physical activity during the pandemic. If there was an increase, it was related to unstructured and outdoor activities. No study has specifically investigated children’s outdoor activity during COVID-19 restrictions. Considering the benefits and importance of outdoor activity, especially in the challenging context of COVID-19, we aim to identify and synthesise the evidence regarding the effects of COVID-19 restrictions on children’s outdoor activities by using systematic review methodology. This review also aims to promote an understanding of the possible impacts on being outdoors in the period of the pandemic and provide direction for future studies.

## 2. Materials and Methods

This review was registered with International Prospective Register of Systematic Reviews (PROSPERO, ref. no. CRD42021289141) and followed the preferred reporting items for systematic review and meta-analysis (PRISMA) statement for systematic reviews to ensure transparency of reporting [22].

### 2.1. Search Strategy

We searched the following electronic bibliographic databases: Education Research Complete, ERIC, MEDLINE, PsycINFO, SPORTDiscus, Psychology and Behavioural Sciences Collection, WHO COVID-19 Database. We used the PICO approach to derive our search terms: Population (Children birth to 12-years), Intervention/interest (COVID-19 restrictions—restrictions here is an umbrella term for lockdown, social distancing and school closure), Comparison (not applicable), Outcome (Outdoor Activity) (see Table 1). Only articles from 2020 onwards were included due to the commencement of COVID-19 restrictions in 2020. Reference lists of identified articles were searched (See Appendix A for detailed search strategy).

### 2.2. Eligibility Criteria

Studies with a quantitative or qualitative design that examined COVID-19 restrictions and children’s outdoor activity were included. In order to identify eligible studies, the Population, Intervention/Interest, Comparison, Outcomes (PICO) framework was applied. Population: For the purpose of the review, children aged from birth to 12 years old were included. A study was included if the participants’ mean age was between birth and 12 years and 11 months. One study [23] had two age groups: 5–11 years and 12–17 years; it was included because it reported the outcomes separately. The age range was selected for this review because outdoor activity is an important direction in early childhood education in COVID-19 [24] where children are aged from birth to 5 years. In addition, children 0–5 and 5–12 years did not receive enough attention compared to adults/older adults [21], but their outdoor activity was also impacted due to the COVID-19 restrictions. Intervention/interest: Studies were included with the following characteristics: (1) the research context was COVID-19; (2) COVID-19 restrictions were in places, such as keeping physical distance, schools, parks, and other entertainment places closed. Outcome: The included studies had at least one outcome related to outdoor activity. The studies were excluded as not clear whether physical activity or play was indoors or outdoors.

### 2.3. Selection Process

EndNote reference software version 20 (Clarivate Analytics, Philadelphia, PA, USA) was used to import study records. Duplicate studies have been deleted. Two independent reviewers screened the articles according to the inclusion and exclusion criteria using Rayyan [25] (software for collaborative systematic review). Selection included two steps: (1) screening of title and abstract, (2) screening of full text. Disagreements were resolved by consensus.

### 2.4. Data Extraction

After selecting included studies, two authors extracted the data in Excel. The name of the first authors, title, research objective(s), study design, country of the study, specific location of the study, period of study, participant characteristic, how impacts were assessed, how other variables were assessed, list of other variables, type of outdoor activity and location, COVID-19 restrictions, outdoor activity findings, and conclusion(s) were extracted from the included studies.

### 2.5. Study Quality Assessment

Two authors measured the quality of the included cross-sectional, cohort, longitudinal, and qualitative studies independently using JBI tools [26]. Disagreements were resolved by discussions (See Appendix A). For longitudinal and cohort studies, the JBI analytical cross-sectional checklist was used based on eight questions to evaluate the quality and risk of bias as it had previously been found to be satisfactory for this purpose [27]. Studies that met 4 or more criteria were considered good/high-quality. The qualitative study checklist included 10 questions, of which a score of 5 or more was considered good/high-quality. Studies with a quality score of at least greater than or equal to the average and using appropriate study design criteria were selected for data extraction. One study [28] included met less than the average of the criteria because its primary outcomes focused on myopia progression rather than outdoor activity, of which we assessed the quality based on the outdoor activity component. We considered the result of the study worthwhile as they reported the outdoor activity outcomes. The JBI tools only have analytical cross-sectional checklist; for descriptive cross-sectional studies, two criteria were not applicable. The descriptive cross-sectional study was assessed without those two criteria.

## 3. Results

### 3.1. Study Selection

Our search identified 349 articles, of which 15 met the inclusion criteria (see Figure 1). Through title and abstract screening, 40 articles were selected for full-text screening. During full-text screening, we sent an email to an author for a full text of an included article. The author provided a separate published article. After the full-text screening of the 41 articles, 26 were excluded. Finally, 15 articles were included for qualitative synthesis.

### 3.2. Study Characteristic

Three longitudinal studies [29,30,31], two cohort studies [28,32], six cross-sectional studies [23,33,34,35,36,37], and four qualitative studies [38,39,40,41] comprised the 15 studies included for qualitative synthesis. Studies were conducted across a wide range of countries with variations in cultural practices, language, and climate (see Figure 2). All included studies investigated the impacts that COVID-19 restrictions had on children’s outdoor activity. Nine studies [29,32,33,34,35,38,39,40,41] reported the outcome as outdoor time/play/activity increased, and six studies [23,28,30,31,36,37] found the opposite results. The majority of participants were aged from 5–12 years old. Regarding the methods applied to measure children’s outdoor activity in the 15 studies, 5 studies [28,29,31,33,37] were conducted using a questionnaire, 5 studies [23,32,34,35,36] used an online survey, 3 studies [38,40,41] were interviews, 1 [30] had both survey and interview, and 1 [39] went for a Froebelian approach to storytelling. Fourteen studies [23,29,30,31,32,33,34,35,36,37,38,39,40,41] included were of moderate-to-high quality and one [28] was low (see Appendix A).

Seven studies investigated the effects of COVID-19 restrictions on young children’s outdoor physical activity combined with associated movement behaviours [32], sedentary behaviour [23,30,36] nutrition [38], screen time [23,30], sleep [23,30,31], and independent mobility [40]. Two studies [33,34] examined how movement behaviours were affected by COVID-19. Two [28,37] evaluated how COVID-19 quarantine affected the development of myopia. Then, the other studies explored children’s experiences of COVID-19 [39], the COVID-19 lockdown impacts on emotional health [35], leisure behaviour and attitudes towards schoolwork [29], and weight-related behaviours of children with obesity [41]. More comprehensive characteristics of the studies are presented in Table 2.

### 3.3. Findings

The systematic review examined the impacts that COVID-19 restrictions had on 0–12 years children’s outdoor activity. Of the 15 studies identified, four [38,39,40,41] were qualitative studies and eleven were quantitative [23,28,29,30,31,32,33,34,35,36,37]. All the qualitative studies reported an increase in outdoor activity during COVID-19 restrictions. Just under half (5/11) of the quantitative studies [29,32,33,34,35] indicated increased outdoor play/activity/time. Studies from Ma et al. [37] and Yum et al.’s [28] main aims were COVID-19 restrictions/quarantine and myopia progression, but our findings only reported the aspect of outdoor activity. Findings from all the studies are presented in Table 3.

#### 3.3.1. Outdoor Activity in Educational Settings

Two qualitative studies [38,39] were conducted in early-childhood education and care centres using semistructured individual interviews and a Froebelian approach to storytelling, respectively. Educators perceived increased outdoor time for children from 2–5 years in response to physical distancing recommendations [38], and the 2–4-year-old children appeared to prefer spending time outdoors at the nursery, and many more chose to be outside now than before COVID [39].

#### 3.3.2. Outdoor Activity at Home Settings

The following studies were conducted when stay-at-home requirements were in place. After four months of lockdown and quarantine measures in China, 7–12-year-old children spent significantly less time on outdoor activities [37]. The Korean government restricted outdoor activities from March 2020, and the times of outdoor activity reduced from 11.8 (h/week) in the pre-COVID-19 period to 6.1 in the post-COVID-19 period in all groups [28]. In Okely et al.’s study [30], the results were from multiple countries, but half of them had strict stay-at-home lockdowns, and on weekdays and weekends, children aged 3 to 5 years spent 81 min and 105 min less time outside, respectively. When required not to leave the house (except for essentials) in Canada, sport and outdoor physical activity saw the greatest reduction [23]. On the contrary, many US families discussed spending more time outside and for longer periods, and discussion and coparticipation were important ways to remain active, even though it was required to not leave the house except for essentials [41].

#### 3.3.3. Outdoor Activity nearby Home/in Public Venues

The significant increase in children’s outdoor activity or outdoor play occurred on the street, sidewalks, and roads in their immediate neighbourhood, the garage, or public venues (such as the park, playgrounds, or outdoor recreation areas) during the COVID-19 outbreak [32,36]. Weekly minutes spent playing outside in public venues showed the greatest percentage difference (95% increase) [32]. On the other hand, the probability of being physically active in a park or trail declined, and performing physical activity in their yard or driveway or on sidewalks and roads outside the neighbourhood did not change [36]. Restrictions imposed by COVID-19 appeared to have a greater effect on girls’ and older children’s physical activity (ages 9 to 13) [36].

#### 3.3.4. Increased Outdoor Hobbies or Activities

During the pandemic, some new outdoor activities or hobbies increased due to families spending more time staying home and being together. A total of 53.1% of parents reported increased outdoor hobbies or activities [34]. Biking, family walking, family hiking, running, and sports (such as basketball on the driveway, road hockey, playing catch, badminton, and soccer) were the major outdoor hobbies/activities mentioned by the parents [23,34].

#### 3.3.5. Families in Outdoor Activities

Family played an important role in children’s outdoor activity during the pandemic. Parental encouragement of physical activity, logistical support (e.g., driving a child to an activity) and involvement in children’s play were associated with being more physically active outdoors. Their encouragement, support, and engagement were also associated with more time devoted to walking and cycling, spending more time playing outside, and overall time being outdoors [34]. Additionally, younger parents, parental cohabitation, having a dog, or those who lived in detached homes with their children spent more time outside [23,34]. Higher ages, fewer children at home, or children from families with a lower socioeconomic status were associated with a lower frequency of outdoor play [29]. One study indicated that unstructured, imaginative, and spontaneous playing increased according to parents and their 7–12-year-old children. The increase featured a greater emphasis on children’s independent mobility to remain active, which led to more time spent playing in the yard and children playing more with siblings [40].

#### 3.3.6. Outdoor Activity and Level of COVID-19 Restriction

There was not enough information to map the studies onto different levels of restrictions [42] and not all studies reported specific levels of restrictions during the pandemic. As a result, this part only focused on studies that reported specific levels of restrictions. Five studies [23,28,36,37,41] were under higher levels of COVID-19 restrictions, such as staying at home except for essential daily exercise or grocery shopping [42]. Among the five, four quantitative studies [23,28,36,37] indicated a decrease in children’s time outdoors, and one qualitative study [41] showed an increase. The increase might be the outcome of staying home and having a flexible day rather than being at school. Two studies [33,38] conducted during the less strict level of restrictions (people recommended to not leave the house or no measures) showed that young children’s outdoor time significantly increased. Five studies [29,30,31,35,40] were categorised as a mixed level of restrictions during the outbreak, which refers to the studies being conducted in both higher and relaxed (less) levels of restrictions. Two studies [30,31] reported that children spent less time outdoors. Conversely, the others [29,35,40] found a significant increase.

#### 3.3.7. Summary of Findings

Only two studies were within preschool settings, two studies were in preschool/school and home mixed settings, and eleven were in the home context. All studies reported an increase in outdoor activity in educational settings. Overall, the systematic review revealed more increased outdoor activity impacted by the COVID-19 restrictions. The locations of children’s outdoor activity were different, but most were near their homes such as the yard, street, roads, and garage, or in the park or playground [32,36]. Quantitative studies with strict stay-at-home requirements indicated a decrease in outdoor activity [23,28,36,37]. In eastern countries such as China and Korea, there were consistent decreases in outdoor activity [28,30,37]. School-aged children experienced a higher decline than preschoolers. More time was spent on digital devices for online learning or attending online e-classes [28,37]. All the qualitative studies also showed consistent findings that children spent more time playing outdoors. Unstructured outdoor playtime increased and children preferred being outside [38,39,40,41]. The sample size did not find consistent patterns. There were studies with a larger sample size that showed decreased outdoor activity, while increased outdoor activity was also reported.

## 4. Discussion

The findings are mixed, with all qualitative studies indicating an increase and more than half of the quantitative studies finding a decrease in outdoor activity. For the qualitative studies, it seems there is a perception of being outdoors more and an interest in more outdoor activity. For the quantitative studies, there is an indication that a decrease in outdoor activity may be in countries with strict stay-at-home restrictions during the pandemic. Unfortunately, many articles did not include information about the time of data collection and types of restrictions relevant to the participants. These details should be included when reporting studies in future. While the overall results were almost evenly split between finding an increase or a decrease, the trend appeared to be that higher levels of restrictions had a negative impact on children’s activity. Lower-level restrictions allowed children and their carers to find options to permit greater outdoor activity. Nonetheless, the inconsistency in findings indicates that more research is needed to determine if the changes detected are attributable to factors other than COVID-19, and suggests that more research across different contexts, especially in countries that were not represented, is needed.

### 4.1. In Educational Settings and at Home

We found different contributions from educational settings and home. The results underscore the importance of both contexts for children’s outdoor activity. In early childhood education and care centres, all the studies reported an increase in outdoor activity by educators or children. Children were taken outside more often than normal and educators saw this as a way to adhere to guidelines and as a relief for them associated with COVID-19 regulation hypervigilance [38]. Children expressed a desire to be outdoors and have extended time to play [39].

The level of restriction appeared to impact contributions of the home environment to outdoor activity. During the COVID-19 period, outdoor activity and time decreased, and the reduction was greater among the most restricted groups (full-lockdown group) [31]. On the other hand, an increase was reported when children were able to play outside at a nearby the house (such as yard, garage, or street), in the neighbourhood park or playground, and on sidewalks and roads [32,36,40]. One investigation found that providing safe recreation programs (e.g., masking, physical distance, and limited utilisation of shared physical facilities) in underused green spaces (such as sports fields, playgrounds, or parks) might encourage outdoor physical activity during COVID-19 outbreaks [43].

There were two studies conducted both in preschools/schools and at home. Children who participated in the study when school was partially closed (half online and half in the classroom) were found to spend significantly less time outdoors [31]. In contrast, another study indicated that throughout the epidemic, Swedish children spent much more time outside on weekdays and weekends. This finding may be related to the fact that schools and childcare facilities remained open and offered children the opportunity to participate in planned activities [33].

### 4.2. Family Demographic, Parental Support, Social and Home Environment Factors

Family demographics such as parents/children’s age, family’s socioeconomic status, and marital status were found to be associated with children’s outdoor activity during the pandemic. Older children or those from lower-socioeconomic-status families were reported to play outdoors less frequently [29]. Previous studies concluded that a greater proportion of young children from low-income families indicated a decrease in outdoor activities [14,44]. Children and youth of younger parents and parents who were cohabiting were significantly associated with more time spent playing outside [34]. Parental support also impacted children’s outdoor activity during the outbreak. Parental encouragement or logistical support of a physical activity, or parent coparticipation, were associated with more outdoor physical activity and outdoor play [23,34].

Social factors (number of siblings, family dog ownership) influenced the time that children spent outdoors at the time of the pandemic. Families with a higher number of children spent more time playing outside and strengthening sibling relationships because they played together more frequently [29]. There were consistent results that children who reported increased outdoor activities were more likely to come from households with multiple children [14,45]. Having a dog encouraged families to get outside and move together, thus spending more time playing outdoors [23,29]. Moreover, home environment factors played a role in children’s outdoor activity within COVID-19 restrictions. Children who lived in detached homes and houses with outdoor spaces spent more time outdoors [30,34]. Recent studies indicated that having an outdoor space at home was a critical factor that contributed for children’s physical activity, as well as having access to outdoor action possibilities at preschool to be shared by peers [44,45]. Hence, our finding is consistent with the existing literature. The results of the review found that access to public green spaces is particularly important in densely populated urban areas, and for those without access to a private garden or yard [30]. Importantly, access to public green space can only occur when restrictions permit activities outside of the home.

### 4.3. Implications for Interventions

This review suggests that children may have more time outdoors during COVID-19 if (a) educators take children outside more to adhere to COVID-19 guidelines; (b) parents encourage, support, and engage in children’s outdoor activity and develop new outdoor hobbies; (c) families, especially those with a lower socioeconomic status, organise outdoor activities in spacious outdoor spaces frequently; (d) communities provide children with various, multiple, and large public facilities such as parks or playgrounds.

### 4.4. Recommendations for Future Researchers

The present review revealed significant gaps in understanding of the impact of COVID-19 restrictions on children’s outdoor activities. The studies reviewed have not had outdoor activity as the primary focus. Despite this, there are some high-quality data available. It will be helpful if researchers continue to include a measure of outdoor activity in their research to further understanding of the impacts of restrictions. It is recommended that researchers aim to use instruments that have been used in the studies included in the present review. Variation in the measures used is a current difficulty. Therefore, the timing of the study, especially concerning the focus on the changes in children’s outdoor activities or levels of the COVID-19 restrictions during the pandemic, needs to be specified to improve the comparability of the studies. Severity of restrictions should also be considered in further studies. Moreover, the outdoor activity details are also worth investigating for future interventions and benefits.

### 4.5. Strengths and Limitations

The current review includes studies with a focused and broader age group. With broad search terms and comprehensive inclusion criteria, we found 15 eligible studies that were independently screened by two reviewers. Fourteen higher-quality (above-average) papers were included, which enhances the validity of the findings.

There were a few limitations of this review. Some studies applied self-reported surveys, which may be at risk for recall bias. We acknowledge that the literature search process was conducted between November 2021 and February 2022, and several studies were excluded due to it being in the publication process/grey literature. Hence, there is also an element of selection bias with current findings. As a result of fewer studies and heterogeneity, there was no meta-analysis. Due to the fact that this review only included articles published in English, it is possible that papers published in other languages were not included.

## 5. Conclusions

The findings from this review suggest that COVID-19 restrictions impact children’s outdoor activity. Both increases and decreases have been reported, and it seems that contextual factors are likely to influence the direction of change. The current evidence indicates that family demographics, social factors, house environment characteristics, access to outdoor spaces, and parental support/encouragement/co-play are vital factors correlated to children’s outdoor activity during the pandemic. Having younger/supportive parents, siblings, a dog, living in detached homes or places with access to outdoor spaces, or a higher-socioeconomic-status family appeared to increase children’s outdoor activity during the COVID-19 period. The evidence also supports the recommendation for educators to increase children’s play time outdoors to adhere to physical distancing guidance and sanitisation requirements. Moreover, limited evidence suggests that contexts where COVID-19 restrictions led to decreased outdoor activity were associated with less peer socialisation. Children expressed desires to be outdoors and play with their friends. In addition to promoting active outdoor play, it is important to prevent the spread of viruses among children. Although there are mixed results, we believe there is sufficient evidence of a detrimental impact of strict lockdowns on young children’s health and development. It is particularly important that all children have the opportunity to spend time outdoors even when high level restrictions are in place. Authorities should provide guidance for families on how to make use of available outdoor space without increasing the risk of contamination. Future interventions and studies can benefit from the findings of the review.

## Figures and Tables

**Figure 1 children-09-01564-f001:**
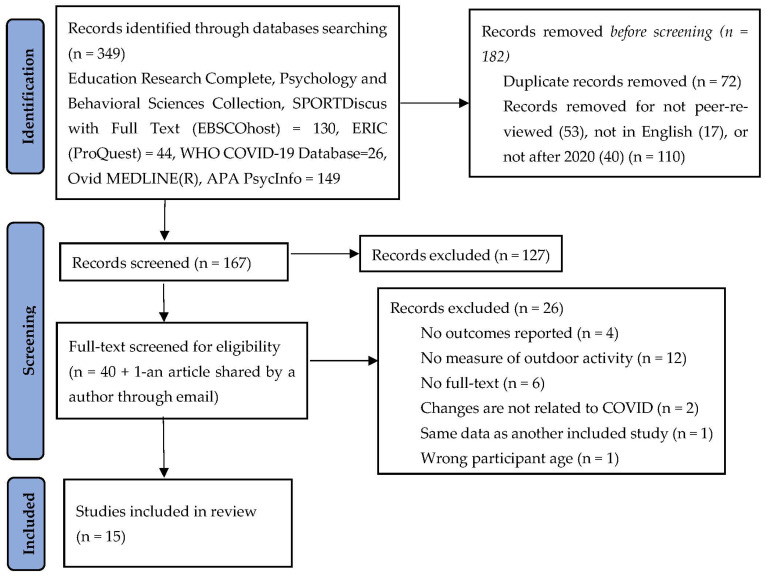
PRISMA flowchart of the study selection process. (PRISMA flow diagram [22]).

**Figure 2 children-09-01564-f002:**
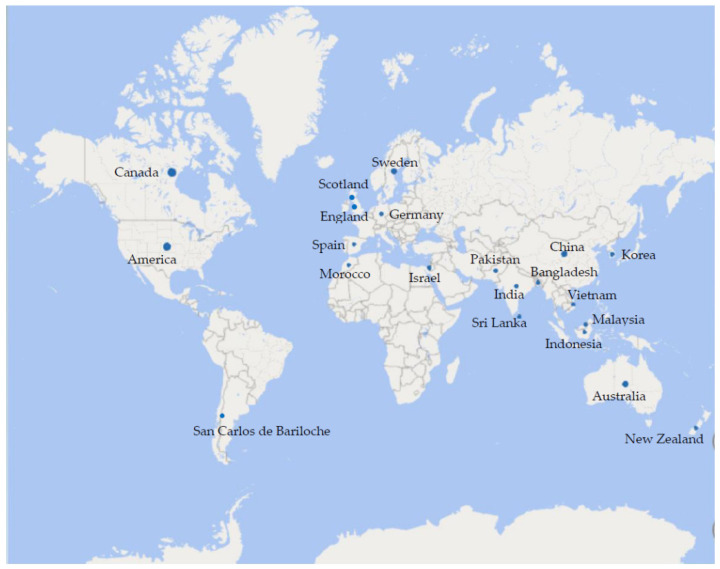
Countries involved in the included studies.

**Table 1 children-09-01564-t001:** Search terms.

PICO	Booleans	Search Terms
Population		“child*” OR “kid” OR “boy” OR “girl” OR “infant” OR “toddler” OR “early years” OR “youth”
Intervention/interest	AND	“COVID-19” OR “coronavirus” OR “COVID-19 pandemic” OR “2019-ncov” OR “SARS-CoV-2” OR “cov-19” OR “pandemic” OR “lockdown” OR “social distanc*” OR “School clos*”
Outcome	AND	(“outdoor” OR “outdoor spaces” OR “outside” OR “nature” OR “park” OR “playground” OR “garden” OR “front yard” OR “green space*” OR “neighborhood” OR “street*”) AND (“play” OR “free play” OR “unstructured play” OR “independent mobilit*” OR “activit*” OR “unorganized physical activit*” OR “organized physical activit*”)
Search strategy example		
Education Research Complete, Psychology and Behavioral Sciences Collection, SPORTDiscus with Full Text (EBSCOhost)	Limited to peer-reviewed; limited after 2020; limited to English	(COVID-19 or coronavirus or “COVID-19 pandemic” or 2019-ncov or SARS-CoV-2 or cov-19 or pandemic or lockdown or “social distanc*” or “School clos*”) AND (outdoor or “outdoor spaces” or outside or nature or park or playground or garden or “front yard” or “green space*” or neighborhood or street*) AND (play or “free play” or “unstructured play” or “independent mobilit*” or activit* or “unorganized physical activit*” or “organized physical activit*”) AND (child* or kid or boy or girl or infant or toddler or “early years” or youth)

* was used for truncation to include all variants of a search term.

**Table 2 children-09-01564-t002:** Study Characteristic.

Author	Study Design	Country	Mean Age and Range (Years)	Sample Size	Outcomes Assessed	Research Aims
Delisle et al., 2020 [33]	cross-sectional	Sweden	During the pandemic 4.0 ± 0.5, 3–5	100	Outdoor time, COVID-19 questionnaire	To examine the impact of the COVID-19 pandemic on preschool children’s movement behaviours and the proportion of them meeting the guidelines and evaluate the viability of the SUNRISE study’s methodology.
Dunton et al., 2020 [36]	cross-sectional	The United States	During the pandemic 8.71, 5–13	211	Outdoor physical activity, online survey	To assess how U.S. children’s (ages 5–13 years) physical activity and sedentary behaviours have been affected by the COVID-19 pandemic during the spring of 2020.
Lafave, L. et al., 2021 [38]	qualitative	Canada	During the pandemic Educator 42.65, children (2–5 years)	17	Outdoor time, semistructured, individual interviews through an online communication platform (Google Meet)	To investigate the perspectives of early childhood educators on the effects of COVID-19 guidelines on nutrition and physical activity practices in their ECEC settings.
Ma et al., 2021 [37]	cross-sectional	China	Pre- and during the pandemic 9.9 ± 1.7, 7.0–12.0	201	Outdoor time, paper questionnaire	To investigate the effect of home quarantine during the COVID-19 pandemic on myopia progression in children and its associated factors.
Moore et al., 2020 [23]	cross-sectional	Canada	During the pandemic Children (8.1, 5–11) youth (14.9, 12–17)	1503	Outdoor activity, online survey (French or English)	To assess how Canadian young children’s physical activity, play, screen use for leisure, and sleep changed using a rapid and large-scale method during the early COVID-19 crisis.
Moore et al., 2021 [34]	cross-sectional	Canada	During the pandemic 11.5, 5–17	T1:1472 T2:1568 Participants in T1 and T2 are a similar sample of parents	Outdoor activity, online survey	To explore young children’s movement behaviours in October 2020 compared with the beginning of COVID-19 in Canada.
Nathan et al., 2021 [32]	retrospective cohort	Australia	During the pandemic 6.9, 5–9	157	Outdoor play, online survey	To investigate the impact of COVID-19 restrictions on Western Australian young 5–9-year-old children’s physical activity behaviour and related movement behaviours.
Okely et al., 2021 [30]	longitudinal	Australia, Bangladesh, China, Hongkong (China), India, Indonesia, Malaysia, Morocco, Pakistan, Spain, Sri Lanka, Sweden, United States, Vietnam	T1 pre- and during the pandemic: 4.4, T2 during the pandemic: 5.2; 3–5	948	Outdoor time, Time1: survey (via interview or self-administered) Time2: telephone interview or online survey	To examine how, compared with the time period pre COVID, the COVID-19 pandemic influenced physical activity, sedentary behaviour, screen time, and sleep among preschoolers. Further, to examine the relationship between COVID-related restriction levels, parent and family factors, and changes in young children’s physical activity, sedentary behaviour, and sleep.
Pascal et al., 2021 [39]	qualitative	England, Scotland, and New Zealand	During the pandemic None, 2–4 years	58	Outdoors, a Froebelian approach to storytelling	To support children to express their narratives about their experiences of the COVID pandemic.
Pelletier et al., 2021 [40]	qualitative	Canada	During the pandemic 9.6, 8–12	45	Outdoor time, interviews	To investigate how children and parents perceive children’s independent mobility and physical activity during the pandemic.
Poulain et al., 2021 [29]	longitudinal	Germany	During the pandemic 5.56, 1.44–10.69	285	Outdoor play, online questionnaires	To investigate the leisure behaviour and the attitudes towards schoolwork in German children during the initial COVID-19-related lockdown, at the start and the end of school closings, respectively.
Schnaiderman et al., 2021 [35]	prospective, descriptive, cross-sectional	San Carlos de Bariloche	During the pandemic 11.1, 6.2–18.1	267	Outdoor activity, online, self-administered survey	To assess the impact of COVID-19 lockdown on the emotional health of children and adolescents attending primary or secondary school.
Shneor et al., 2021 [31]	longitudinal	Israel	Pre-pandemic 10.2 ± 0.9, during the pandemic 11.5 ± 0.9, 8–12	19	Outdoor time, questionnaires	To examine how COVID-19 restrictions affected 19 boys’ physical activity, outdoor time, and sleep before and during the outbreak and after the pandemic social restrictions were lifted by applying objective measures.
Yum et al., 2021 [28]	retrospective cohort	Korea	Pre- and during the pandemic, 10.1 ± 2.5, 5–15	103	Outdoor activity, questionnaires	To assess how COVID-19 restrictions affected the development of myopia.
Neshteruk et al., 2021 [41]	qualitative	America	During the pandemic 9.7 (±2.8), 5–17	51	Outdoor time, interviews using secure phone or video conferencing software.	To explore changes in the weight-associated behaviours in obese children after the pandemic.

**Table 3 children-09-01564-t003:** Findings from studies.

Outcome	Data Collection Time	Reference	Context	Outdoor Activity Findings
Decreased outdoor activity	Approximately 1 month after the WHO declared COVID-19 a global pandemic, 2020	Moore et al., 2020 [23]	Home in Canada with parents	The most dramatic decrease occurred in outdoor physical activity and sport (2.28/5.00 and 1.96/5.00 for children and youth, respectively). 22.7% reported increased outside hobbies or activities. There was a correlation between parents’ marital status (cohabitation) and more outdoor play among children (0.10). Children that lived in detached homes were more physically active outdoors (0.12) and spent more time walking and cycling (0.13). Dog ownership was correlated with more outside play (0.11).
	Pre-COVID-19: January 2019 to May 2019, March 2020. Post-COVID-19: March 2020, January 2021 to March 2021.	Yum et al., 2021 [28]	Home in Korea with parents	The times of outdoor activity (h/week) post-COVID-19 (6.1) were significantly lower than pre-COVID-19 (11.8) in all groups (all *p* < 0.05).
	25 April–16 May 2020.	Dunton et al., 2020 [36]	Home in the U.S. with a parent or legal guardian	Parents of older children (ages 9–13) compared to younger children (ages 5–8) perceived greater decreases in physical activity between the pre-COVID and the onset of COVID-19 (*p* = 0.003). Increase in the possibility of engaging physical activity at home, in the garage (OR = 2.49, 95% CI [1.35, 4.60], Wald = 8.593, *p* = 0.003), and on the sidewalks and roads in their neighbourhood (OR = 1.92, 95% CI [1.04,4.60], Wald = 4.28, *p* = 0.038) from before COVID-19 to the beginning of the COVID-19 period after controlling for child sex, child age group, child ethnicity (Hispanic vs. non-Hispanic), parent employment status (full-time vs. part-time), parent marital status (married vs. not married), and annual household income. The likelihood of being physically active on a park or trail (OR = 0.47, 95% CI [0.23, 0.97], Wald = 4.22, *p* = 0.040) reduced from the period prior to COVID-19 to the start of the COVID-19 period. The likelihood of children engaging physical activity in their yard or driveway (OR = 1.32,95% CI [0.76, 2.31], Wald = 0.95, *p* = 0.329) or on sidewalks and roads outside the neighbourhood (OR = 0.76, 95% CI [0.25,2.33], Wald = 0.288, *p* = 0.633) did not change.
	April 2019 and March 2020, May–June 2020	Okely et al., 2021 [30]	In urban and rural areas, parents were recruited through early-childhood education and care (ECEC) services and villages in Australia, Bangladesh, China, Hongkong (China), India, Indonesia, Malaysia, Morocco, Pakistan, Spain, Sri Lanka, Sweden, United States, Vietnam	On weekdays and weekends, children spent 81 min (*p* = 0.021) and 105 min (*p* = 0.003) less time outside, respectively.
	Pre-pandemic: June 2019 to March 2020. During social restrictions: November 2020 to March 2021. Post- restrictions: April to June 2021	Shneor et al., 2021 [31]	All schooling was online (9 children); when school was partially online and partially classroom based (10 children)	Time spent outside significantly decreased from 1.8 ± 0.8 h (pre-pandemic, *p* = 0.002) to 0.5 ± 0.3 h (during restrictions), and then went back to 1.6 ± 0.6 h after the restrictions were lifted. (post-restrictions vs. pandemic restrictions *p* = 0.001; post-restrictions vs. pre-pandemic *p* > 0.99).
	April and May 2019, October and November 2019, May 2020	Ma et al., 2021 [37]	Home in China with parents/caregivers	Parents/caregivers indicated that children spent much less time outside during visit 2 (0.49 ± 0.23 h/d; *p* < 0.001) than they did during visit 1 (1.11 ± 0.35 h/d, *p* < 0.001), showing a dramatic decline in outdoor time from February 2020.
Increased outdoor activity	March and May of 2019, May/June 2020	Delisle et al., 2020 [33]	Preschools and home in Sweden with parents	Swedish children spent significantly more time outside during the COVID-19 pandemic on weekdays (+124 min/day) and weekends (+124 min/day; all *p*-values ≤ 0.001).
	July to August 2020	Lafave, L.et al., 2021 [38]	Early-childhood education and care centres with educators in Canada	Educators perceived more time spent outdoors and more physical activity time for children.
	Pre-COVID-19: April 2020. Post-COVID-19: October 2020.	Moore et al., 2021 [34]	Home in Canada with parents	53.1% of parents reported increased outdoor hobbies or activities. Children and youth with younger parents have more time to spend outdoors (r = −0.12). Children and youth living in detached homes devoted more time outdoors (r = 0.12), walking and biking outdoors (r = 0.14), more time playing outdoors (r = 0.16), and overall time being outside (r = 0.13). Parents who encouraged physical activity were associated with being more physically active (r = 0.17), time spent walking and biking (r = 0.17), time playing outdoors (r = 0.15), and spending more time outdoors overall (r = 0.13). Parental involvement in children’s play was associated with being more physically active outdoors (r = 0.29), more walking and biking (r = 0.32), more time playing outside (r = 0.40), and overall time being outdoors (r = 0.36). Logistical parental support for physical activity, such as driving children to activities, was associated with being more physically active outdoors (r = 0.20), more walking and cycling (r = 0.11), and more time playing outside (r = 0.13), and more overall time being outdoors (r = 0.17).
	15 May to 5 June 2020.	Nathan et al., 2021 [32]	Home in Australia with parents	Time playing outdoors around the house (yard or street), in public venues (park or playground), or in outdoor recreation areas (with a 95% increase, the greatest percentage difference found for weekly minutes) indicated a significant increase from pre-COVID to keeping distances during COVID-19 period.
	Unclear	Pascal et al., 2021 [39]	Early-childhood education and care centres in England, Scotland and New Zealand with practitioners	Practitioners in all settings have reported that the children appeared to prefer being in outdoor spaces when at nursery, with many more choosing to spend time outside at the time of the study than before COVID.
	September–December 2020	Pelletier et al., 2021 [40]	Home in small urban and rural areas in Canada with parents or guardians	Unstructured, imaginative, and unscheduled playtime increased as perceived by parents and children, including an increased dependency on children’s independent mobility to remain active, resulting in more time playing outdoors in the yard.
	March/April 2020, t1, April/May 2020, t2	Poulain et al., 2021 [29]	Home in Germany with parents	The frequency of children playing outdoors increased from 60% to 71%. Compared to 68% of parents in high-socioeconomic-status families, 51% of parents in low/middle socioeconomic status families reported that their child played outdoor daily. In single-child households, 54% of children were reported to play outdoors every day, compared to 70% in three-child households.
	September and October 2020	Schnaiderman et al., 2021 [35]	Home in Argentina with parents	Being outdoors increased from 22.1% in April (when there was complete lockdown) to 74.2% in August (People were permitted to walk and cycle in the neighbourhood in June; and the ski resorts, bars, and restaurants reopened in July.).
	April to June 2020	Neshteruk et al., 2021 [41]	Home in the U.S. with parents	Due to staying at home and having fewer organised school days, several of these families discussed spending more time outside. Families with less active children mentioned the inability to go outside as a factor in their inactivity (For instance, absence of parental supervision or safety concerns).

## Data Availability

All data generated during this study are included in this published article [and its Appendix A].

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
