# Peer review of "Impacts of COVID-19 Restrictions on Young Children’s Outdoor Activity: A Systematic Review"

_children, 2022, doi:10.3390/children9101564_

Round 1
Reviewer 1 Report
This work is a systematic review of the literature, it meets all the points recommended by PRISMA, which facilitates its evaluation and better understanding, I only have some minor suggestions that could help the article report.
The flowchart can be improved so that it is more attractive to readers, I suggest using the same format suggested by PRISMA.
In the case of the search criteria, it was a good idea to put them in a box, but in addition to the criteria, you should put the search strategy of at least one of the databases used, PRISMA suggests the Medline strategy. In addition to the above, as an extra suggestion, I suggest the authors to place all the search strategies for each database as a supplementary file, this would help better reporting and knowledge to other authors of what is the correct way to create search strategies in the different databases.
In table 3, there seems to be an editing error at the end of the table, please correct it.
Author Response
Thank you for your time and valuable feedback.
|
Comment |
Response |
Page number/ location |
|
The flowchart can be improved so that it is more attractive to readers, I suggest using the same format suggested by PRISMA. |
We have changed the flowchart and used the same format suggested by PRISMA. |
Figure 1, Page 4, line 144-173 |
|
In the case of the search criteria, it was a good idea to put them in a box, but in addition to the criteria, you should put the search strategy of at least one of the databases used, PRISMA suggests the Medline strategy. In addition to the above, as an extra suggestion, I suggest the authors to place all the search strategies for each database as a supplementary file, this would help better reporting and knowledge to other authors of what is the correct way to create search strategies in the different databases. |
We added one supplementary document for detailed search strategies. |
Supplementary file, Page 2-3, line 84-87 |
|
In table 3, there seems to be an editing error at the end of the table, please correct it. |
We corrected the editing error at the end of table 3. We removed the repetitive information on the left column about the the Neshteruk et al. |
Table 3, Page 11-12, at the end of the table |

Reviewer 2 Report
Work in the field of sociology.
The review analysis is technically well done and its coverage is very detailed. Despite the heterogeneity of the research results, the conclusions of the review are systematized and generalized.
The work systematizes the issue, despite the fact that the number and variety of often contradictory data make it difficult to draw unambiguous conclusions.
For consideration: wouldn't the headline "Implications for Future Research" be replaced with "Recommendations for Future Researchers"?
Author Response
Thank you for your time and valuable feedback.
|
Comment |
Response |
Page number/ location |
|
For consideration: wouldn't the headline "Implications for Future Research" be replaced with "Recommendations for Future Researchers"? |
We have changed the heading to “Recommendations for Future Researchers.” |
Page 14, line 371 |

Reviewer 3 Report
your study shows interesting aspects of pandemic and post pandemic period of time, but for the systematic rewiew to be published you need to take your time to do important changes for publication. I wish you all the best.
Systematic rewiew
is an article with an impact on the target age group. it is important to know these factors involved in the lives of children during the pandemic and post-pandemic period. for starters, I would ask you to simplify the expression for the inclusion and exclusion criteria, so as not to lose readers. please reduce the text on lines 86-107 and 141-147. The flowchart explains very well the selection process.
Line 223-236
237-251
252-261
please, let's write them more concisely, and narrow down the text a bit for the 3 paragraphs.
Line 268
Table 3 as the results appear, they are interesting, but I would prefer that you group them by periods (time interval) and simplify the table, because it is difficult to follow and loses the scientific interest and the message conveyed. as a particularity, I would add here a graph that can highlight everything you described.
I would like to find in this review, a demographic table with the type of residence, food, on each age group, so that I can see exactly how these changes in daily life took place and especially how they affected the quality of life.
It is necessary to show a graphic report for differences in changes by age group.
The study will help determine the amount of family health behaviour changes brought on by COVID-19 restrictions and will serve as a decision-making tool for measures needed to address negative changes in order to lower the likelihood of long-term health repercussions/consequences.
Line 320-334
it is necessary to support what you explain here, in the results part with the statistics that show what you describe to us.
Line 368-384
the fact that it does not bring anything new as information, makes me ask you to rethink the review to be able to convey that message beyond the one that already emerges from the published studies and from everyday life.
Author Response
Thank you for your time and valuable feedback.
|
Comment |
Response |
Page number/ location |
|
Systematic rewiew is an article with an impact on the target age group. it is important to know these factors involved in the lives of children during the pandemic and post-pandemic period. for starters, I would ask you to simplify the expression for the inclusion and exclusion criteria, so as not to lose readers. please reduce the text on lines 86-107 and 141-147. The flowchart explains very well the selection process. |
We reduced the text on lines 86-107, and 141-147.
|
Page 3, line 89-91; Page 4, line 140-143 |
|
Line 223-236, 237-251, 252-261 please, let's write them more concisely, and narrow down the text a bit for the 3 paragraphs. |
We agree, these paragraphs could be more concise. We have reduced the text. |
Page 8, line 241-246, 253-255, 270-271 |
|
Table 3 as the results appear, they are interesting, but I would prefer that you group them by periods (time interval) and simplify the table, because it is difficult to follow and loses the scientific interest and the message conveyed. as a particularity, I would add here a graph that can highlight everything you described. |
Thank you for this point. We agree, it would be ideal to include the time interval. We tried to check the time of the data collection and map against the time in the pandemic, but there isn't enough information in the papers to report this in Table 3. Instead, we added one row to show the data collection time as reported. Additionally, we acknowledged this in page 13, line 379-382, and recommended the need for further studies to specify timing in data collection to enable comparison.
|
Table 3, page 9, line 295
|
|
I would like to find in this review, a demographic table with the type of residence, food, on each age group, so that I can see exactly how these changes in daily life took place and especially how they affected the quality of life. |
The fact that the context of the studies focuses on restrictions in home setting or school setting, there was no sufficient information to categories the studies into the suggested demographics, other than the ones we reported on table 2. Additionally, we want to ensure the review is within its scope/aim. We were therefore, unable to make the change, but we appreciate your comment. |
Table 2, Page 5, line 201 |
|
Line 320-334 it is necessary to support what you explain here, in the results part with the statistics that show what you describe to us. |
Thank you for pointing out this. We added the statistical significance. |
Table 3, Page 9, The outdoor activity findings of Dunton et al. |
|
Line 368-384 the fact that it does not bring anything new as information, makes me ask you to rethink the review to be able to convey that message beyond the one that already emerges from the published studies and from everyday life. |
Thank you for the comment. The new information is that there are mixed findings with respect to impact of covid-19 restriction on children. Our additional message is the recommendation for further studies to consider more detailed changes in children's outdoor activities or based on levels of the COVID-19 restrictions. We have made this message clearer. |
Page 13, line 379-382 |

Round 2
Reviewer 3 Report
Dear authors, in order to be published, your work still requires some significant changes. please follow the recommendations and return to the important aspects made to you,
best regards.
Second review,
Please, try to make the abstract more concise in detail, and include the number of people/children.
Complete the conclusions with the proposed strategy, something that suggests the completion of the activities carried out outside, in the event of a similar COVID-19 situation. probably the implementation of a curriculum for such situations, in the future, would be of great interest. especially in pediatric populations.
these studies selected by you must be compared and I would definitely like to see a contingency table, for demographic comparisons, (categories age, gender, regions), to make frequency comparisons. (it is easy to understand that during the pandemic the sessions of sports activities, etc. have decreased). but I want to see the impact on the population categories in the studies presented here.
You talked about myopia, please show me how we can prove this, related to the time spent in front of the screen probably and statistics. do we have an ophthalmology consultation somewhere? Please review the article, it is necessary to clarify the mentioned aspects and improve the message.
please touch on these important aspects to complete a systematic review that will be of interest and especially to extract the most significant aspects of the COVID-19 pandemic for the future
Author Response
Response to Reviewer 3_ Round 2
Thank you for your time and valuable feedback.
|
Comment |
Response |
Page number/ location |
|
Please, try to make the abstract more concise in detail, and include the number of people/ children. |
We have revised the abstract to make it more concise.
We appreciate the comment to include the overall number of participants. We have not included the number of participants as it is not the convention for systematic reviews that include multiple methods. The convention is to report the number of studies that are qualitative, quantitative or mixed methods as we have done. We checked some recent systematic reviews and found that this is the convention that is followed for publications in top-tier journals e.g. https://www.ncbi.nlm.nih.gov/pmc/articles/PMC7521017/ https://gh.bmj.com/content/6/4/e004377.abstract https://www.ncbi.nlm.nih.gov/pmc/articles/PMC7613039/ https://www.mdpi.com/1660-4601/15/4/590 Additionally, due to the diversity of the methods used and the types of participants, we do not believe it will be meaningful to provide an overall number of participants. We hope this explanation covers our reason for not making this recommended correction. |
Page 1, line 10-23 |
|
Complete the conclusions with the proposed strategy, something that suggests the completion of the activities carried out outside, in the event of a similar COVID-19 situation. probably the implementation of a curriculum for such situations, in the future, would be of great interest. especially in pediatric populations. |
Thank you for the suggestion. We completed the conclusion with suggestions for highlighting outdoor activity for children during the pandemic. We also added one more recommendation for future researchers since there are only three studies included that mentioned the outdoor activity details. |
Page 13, line 381-383 Page 14, line 412-417
|
|
these studies selected by you must be compared and I would definitely like to see a contingency table, for demographic comparisons, (categories age, gender, regions), to make frequency comparisons. (it is easy to understand that during the pandemic the sessions of sports activities, etc. have decreased). but I want to see the impact on the population categories in the studies presented here. |
We appreciate this comment and we too would like to be able to present this type of information. Unfortunately, the studies were too heterogeneous for us to present meaningful comparisons. Ideally we would be able to present a table that includes severity of restrictions, age, gender, SES, dwelling. We were not able to extract all of the information required to do so, and we have a mixture of methods as well as types of participants. If more studies are published, it may be possible to do this in the future, but it was not possible with the present studies. Likewise, we were unable to run any meta-analyses. |
None |
|
You talked about myopia, please show me how we can prove this, related to the time spent in front of the screen probably and statistics. do we have an ophthalmology consultation somewhere? Please review the article, it is necessary to clarify the mentioned aspects and improve the message. |
We decided to remove reference to myopia. Our interest in this work was to extract the information on outdoor physical activity. The studies do not clearly establish a link between reduced outdoor activity and myopia progression. The reviewer question has alerted us to the possibility that our inclusion of reference to myopia may be misleading. We appreciate this comment. The relationship to myopia in one study (Ma et al) seems to be more closely linked to increased online time than to reduced outdoor activity. In the other study (Yum et al), the statistical analysis does not show a clear connection between myopia progression and reduced outdoor activity. We believe it is better to focus on the outdoor activity measures, which are clear, and not associate these with myopia progression, which could be misleading. |
Page 1, line 21-22 Page 4, line 130-131 Page 7, line 206-208 Page 14, line 409-410 |

Round 3
Reviewer 3 Report
dear authors, try to extract those significant changes to express characteristics of each country/continent, about the degree of COVID-19 changes. please find the main idea as the originality of your work.

Author Response
Dear reviewer,
Thank you so much for your time and valuable comments. Please see the attachment.
Kind regards
Authors
